# On the State of Graphene Oxide Nanosheet in a Polyurethane Matrix

**DOI:** 10.3390/nano13030553

**Published:** 2023-01-30

**Authors:** Sergey A. Baskakov, Yulia V. Baskakova, Eugene N. Kabachkov, Elizaveta V. Dvoretskaya, Svetlana S. Krasnikova, Vitaly I. Korepanov, Alexandre Michtchenko, Yury M. Shulga

**Affiliations:** 1Federal Research Center of Problem of Chemical Physics and Medicinal Chemistry, Russian Academy of Sciences, 142432 Moscow, Russia; 2Institute of Microelectronics Technology and High Purity Materials, Russian Academy of Sciences, 142432 Moscow, Russia; 3Instituto Politécnico Nacional, SEPI-ESIME-Zacatenco, Av. IPN S/N, Ed.5, 3-r piso, Ciudad de México 07738, Mexico

**Keywords:** graphene oxide, nanosheet, polyurethane, composite, XPS spectra, Raman spectra

## Abstract

Thermally stable films were obtained from a water-based polyurethane (PU) dispersion with small (0.1–1.5 wt.%) additions of graphene oxide (GO). The films were studied through elemental analysis, X-ray photoelectron spectroscopy, differential thermogravimetry, and Raman spectroscopy. It was found that the introduction of GO into a PU matrix was accompanied by a partial reduction in graphene oxide nanosheet and an increase in the concentration of defects in GO structure. It has been also established that the [C/N]_at_ ratio in the near-surface layer of PU/GO composite films grows with an increase in the content of graphene oxide in the composite films.

## 1. Introduction

Polyurethanes (PUs) are used for manufacturing sealing products, protective abrasion-resistant coatings, elastic molds of decorative elements, paints and varnishes, sealants, machine parts and machine tools, and various rubber products for both household and industrial use [1]. PU is distinguished from other polymeric materials by its high mechanical strength, wide range of elasticity, and excellent impact–viscosity characteristics. In addition, the properties of PU are quite easy to control by changing the chemical structure of isocyanate and polyol through the introduction of fillers [2,3]. In principle, polymer composite materials with small additions of graphene-like materials demonstrate a significant improvement in mechanical and physicochemical properties and performance characteristics, which usually cannot be achieved using pure polymers [4,5,6,7].

In our previous studies, we described a method for obtaining aqueous mixtures of PU with a GO suspension that was stable over time [8]. We obtained mixtures with a GO content of 0.1–2.0 wt.%, which were used to obtain nanocomposite PU/GO films. The nanocomposites demonstrated higher thermal stability and increased mechanical strength compared to the original polymer. It has been established that the introduction of 2 wt.% graphene oxide increases the Young’s modulus of films almost six times. 

In addition to enhancing thermal stability and improving mechanical properties, GO additives are used in composite adsorbents for water purification from dyes [9,10], oils and organic solvents [11,12]. Graphene materials are also used in various sensors [13,14,15,16,17] and in low-frequency energy converters [18].

Interestingly, we did not find the effect of graphene oxide additives on the IR spectrum of composites [8]. Moreover, the same results were obtained earlier [19]. Thus, a paradoxical situation arose where there was no direct information regarding the state of the additive that significantly affected the mechanical properties of the studied samples. In the present study, our goal was to study the Raman spectra of the designated composites given that Raman spectroscopy is a well-known and highly informative method for studying carbon materials, including graphene-like ones [20,21,22]. Another goal of this study was to study the effect of the GO addition on the surface segregation of mobile PU segments in the PU/GO composite by using X-ray photoelectron spectroscopy. As is known from the literature, PU is characterized by a significant difference in the composition of near-surface and bulk grains [23,24,25,26,27,28].

## 2. Materials and Methods

Aliphatic polyurethane dispersion Bayhydrol^®^ UH 340/1 (manufacturer: COVESTRO, Leverkusen, Germany) was used as a source of PU. This material is used as a binder in the formulation of highly elastic waterborne compounds for wood, metal, and plastic, along with being used as a combined binder to improve the elasticity and flexibility of coatings.

Graphite oxide was obtained by the modified Hummers’ method according to the procedure described in [29]. Pencil graphite grade GK-1 was used as a raw material, and the particle size for this brand of graphite did not exceed 30 microns. When graphite is oxidized, the particle size along the basal plane does not increase. Therefore, a GO suspension of a given concentration was prepared through the ultrasonic treatment of a graphite oxide suspension followed by centrifugation at 3000× *g* to remove large nonseparated particles.

The composite PU/GO films were prepared as follows: The calculated volume of the GO suspension was added dropwise to the PU dispersion with stirring. After the introduction of GO, the stirring was continued for 10 min. Then, the mixture was poured into a mold, which was a glass plate, and edged in order to prevent the mixture from spreading. Afterward, the mold was preliminarily leveled on a bar level to obtain a film without a significant thickness deviation. After the mixture had dried, the film was separated from the mold, and the necessary tests were carried out.

The elemental analysis of the samples preliminarily degassed in an argon flow at a temperature of 80 °C for 30 min was performed using a Vario Micro cube CHNS analyzer (Elementar GmbH, Hanau, Germany).

Thermogravimetric analysis (TG) of the samples was performed using an STA 449 F3 Jupiter instrument (Netzsch Geratebau GmbH, Selb, Germany). In order to calibrate the balance, the chamber of the instrument was evacuated (10^−2^ bar) and filled with grade 6.0 He gas (99.9999%). After that, two empty corundum (Al_2_O_3_) crucibles were placed on the holder in the working chamber of the device, and the baseline was recorded. Then, a sample was placed in one of the empty crucibles, and the instrument chamber was once again evacuated and filled with He. The measurements were carried out in the temperature range of 20–400 °C and at a rate of 10 °C/min in a He flow of 50 mL/min.

The Raman spectra were obtained on a Bruker Senterra micro-Raman instrument. The laser excitation wavelength was 532 nm, the power at the measurement point was 1 mW for PU and 0.1 mW for the composites with GO; the beam diameter was ~2 μm. For all samples, 4 spectra at different points were taken (no difference between these points was found); acquisition time was 2 × 30 s.

XPS spectra were obtained using a Specs PHOIBOS 150 MCD9 electronic spectrometer for chemical analysis. When recording spectra, the vacuum in the spectrometer chamber, which is an X-ray tube with a magnesium anode (Mg K_α_ radiation 1253.6 eV), did not exceed 2 × 10^–10^ Torr, and the source power was 225 W. A low-resolution survey spectrum was recorded in the range of 0–1000 eV, and the spectra were recorded in the constant transmission energy mode (i.e., 40 eV for the survey spectrum and 10 eV for individual lines). The overview spectrum was recorded with a step of 1 eV, while the spectra of individual lines were recorded with a step of 0.05 eV.

## 3. Results

### 3.1. Elemental Analysis

Table 1 presents the results of the elemental composition of the studied samples. All the samples of the composite were close in composition to the composition of the matrix. As graphene oxide is made up of the same elements as PU, a small addition of graphene oxide had practically no effect on the composition of the composite, which was determined by the aforementioned method.

### 3.2. XPS Spectra

The results obtained after analyzing the composition of the samples by using the XPS method (Table 2) differ from those obtained by elemental analysis. First, silicon’s presence in the studied samples should attract attention. According to the literature, silicon is often derived from dimethylsiloxane-based contaminants [25,30], and the introduction of GO slightly reduces the surface content of silicon. However, the most striking difference between the composition of the near-surface layers and the volume can be seen in Table 3, which presents the C/O and C/N ratios for the samples under study. It can be observed that the surface C/O ratio was almost two times higher than the volume ratio for both the initial PU and the composites. In principle, surface segregation in PUs is a well-known phenomenon (for example, see [23,24,25,26,27,28,31,32,33,34]). In the case of our study, the surface C/N ratio changed symbatically with the change in the GO concentration, while the volume ratio remained almost constant (Figure 1). Moreover, since nitrogen is bound to urethane and urea bonds (i.e., hard segments), the elemental composition data demonstrated a significant increase in the nitrogen concentration from the surface to the volume, which suggests that the content of urethane and urea near the surface is much less than in bulk. Thus, our findings are consistent with the data obtained by [23,24,25,26,27,28,34].

The XPS spectra of high-energy resolution from PU and PU/GO films can be seen in Figure 2. The decomposition of the spectra was carried out following the algorithm proposed in [35]. The assignment, position, and relative intensities of individual peaks (within a separate spectrum) are provided in Table 4. It can be observed that in the N1s spectrum, along with isocyanate groups (peak at 399.9 eV), a peak with a binding energy of 401.3 eV appeared in the composite, which could be attributed to oxidized nitrogen. In [25], nitrogen with such a binding energy was designated as C–NH_3_^+^. In principle, it is known that graphene oxide interacts with isocyanates [36]. Moreover, it was established by the XPS method that this is a very complex interaction. Thus, the intensity of the peak at 286.55 eV in the spectrum of C1s GO sharply decreased after treatment with isocyanate, while, on the contrary, the intensity of the peak at 287.1 eV increased. This change might have been due to the formation of a carbamate ester instead of hydroxyl, which reduces the number of hydroxyl groups and, accordingly, increases the number of carboxyl groups. In general, the C1s spectrum of graphene oxide treated with isocyanate indicates a partial reduction in graphene oxide because of treatment [37]. Therefore, the data obtained in our study are in full agreement with this scheme, i.e., the sum of the intensities of the peaks at 286.0 and 286.6 eV was higher in the spectrum of the composite (36.68%) than in the spectrum of the original PU (35.34%).

### 3.3. DTG Curves

The TG curves were presented in ref [8], and it was noted that the introduction of GO increased the thermal stability of films based on PU. However, no quantitative estimates of this increase were given. Provided below are DTG curves for a pure PU film and a composite film with a GO content of 1.0 wt.% (Figure 3). It can be observed from Figure 3 that the temperature of the maximum composite degradation rate (T_max_) significantly exceeded that of a pure PU film. Moreover, the numerical values of T_max_ for the studied samples are presented in Table 2. It was found that the T_max_ of all the studied composites was more significant than that of pure PU. However, instead of the expected monotonic dependence of this parameter on the GO content, a dependence with a maximum can be observed, which we cannot yet explain. Comparison of our data on the effect of additives on the thermal stability of polyurethane with data from other authors shows that graphene oxide is a very effective small additive (Table 5). Thus, the addition of only 0.1% GO increases T_max_ by 7.5 °C, which exceeds this value for other additives.

### 3.4. Raman Spectra

The Raman spectra of GO and rGO are well known (for example, see [44,45,46,47]). At the same time, the information regarding the Raman spectra obtained by different authors differs somewhat. In our study, we followed the authors of [48], according to whom the Raman spectra of rGO obtained upon excitation with radiation at a wavelength of 532 nm can be used to obtain data on the oxygen content of crystallinity as well as the degree of disorder in it. Figure 4 shows the Raman spectrum of graphene oxide used in this work. Table 5 shows the position, relative intensities, and half-widths of individual peaks in this spectrum, as well as similar parameters of the spectrum of graphene oxide (for the sake of comparison), which was reduced (i.e., treated with hydrazine vapor). It can be observed from Table 5 that after reduction, the main peaks (D and G) narrowed and the I_D_/I_G_ - ratio increased. Similar changes in the Raman spectra of reduced GO were previously observed in [49].

Figure 5 presents the Raman spectra of the polymer and its composites with GO. In the case of the spectrum of pure PU with the range of 50–3500 cm^−1^, there were practically no intense peaks. Moreover, the introduction of 1% graphene oxide led to the appearance of peaks characteristic of graphite-like structures in the spectrum. Furthermore, when 1.5% graphene oxide was introduced, the peaks characteristic of the polymer practically does not appear in the spectrum. The results of the decomposition of the PU/GO spectra in the region of 750–2000 cm^−1^ are also shown in Table 6, which shows that, as in the case of restoration, the main peaks (i.e., D and G) in the spectra of the composites had a smaller half-width compared to that in the initial graphene oxide. Additionally, the I_D_/I_G_ ratio in the spectra of the composite increased, and this increase was much higher than the increase in the reduction in GO. Consequently, new defects appeared in the graphene-like structure of GO. Such defects can be chemical bonds with the PU matrix. 

## 4. Discussion

As it was previously reported [8], introducing a small amount of GO (i.e., up to 2 wt.%) effectively strengthened the PU matrix. Moreover, the Young’s modulus increased symbatically with the increasing concentration of the additive, reaching a value of 42.95 MPa, which was almost six times higher than that of the initial PU. In the literature, this behavior is logically associated with physical crosslinking between the rigid domains of the PU matrix and the functional groups on the GO nanosheet surface due to the formation of hydrogen bonds [50,51,52,53,54]. However, IR spectroscopy, which is one of the most effective methods for studying hydrogen bonds, turned out to be insensitive in our case [8]. Nevertheless, as is evident in this work, the study of composites by the Raman method made it possible to establish that GO nanosheets embedded in a PU matrix are partially reduced due to the chemical interaction with the functional groups of the matrix. Such an interaction may result in the appearance of such groups as C–NH_3_^+^, which XPS detected.

## 5. Conclusions

The PU/GO films with a GO content of 0.1–1.5 wt.% were studied with the help of elemental analysis, X-ray photoelectron spectroscopy, differential thermogravimetry, and Raman spectroscopy. From this study, the following conclusions can be drawn:The comparison between the data on the volume and the surface elemental composition reveals a significant increase in the nitrogen concentration from the surface to the volume, which suggests that the content of urethane and urea near the surface is much less than in the bulk.It has been established that the [C/N]_at_ ratio in the near-surface layer of PU/GO composite films grows symbatically with an increase in the content of graphene oxide in the composite.In the N1s spectrum of the composite, along with the peak from the isocyanate groups (399.9 eV), another peak appeared with a binding energy of 401.3 eV, which can be attributed to oxidized nitrogen.The peak intensity in the C1s spectrum underwent a slight decrease due to the hydroxyl and peroxide groups in the composite compared to the initial PU.The temperatures of the maximum degradation rate T_max_ of all the studied composites were higher than that of pure PU. The value can increase by more than 26 °C with the introduction of only 0.5 wt.% GO.The change in the Raman spectra indicates that the formation of the composite is accompanied by a partial reduction in GO nanosheet and an increase in the concentration of defects in its structure.

## Figures and Tables

**Figure 1 nanomaterials-13-00553-f001:**
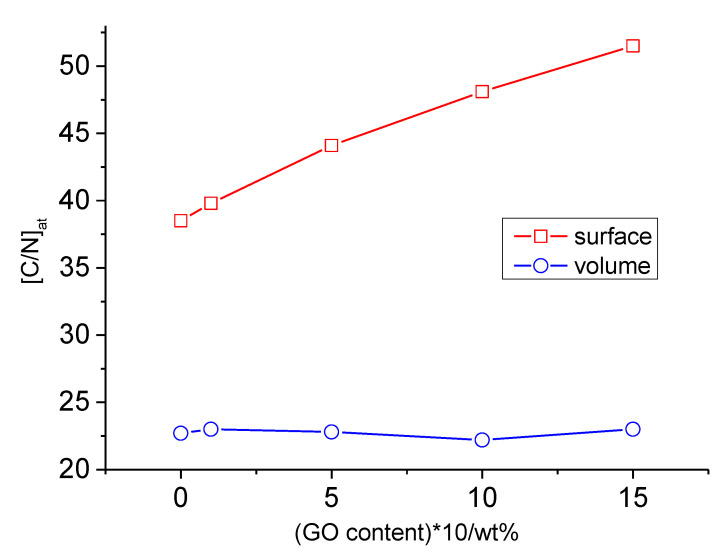
Dependence of the surface and volume ratio [C/N]_at_ on the content of graphene oxide in the PU/GO composite (* content of graphene oxide multiplied by 10).

**Figure 2 nanomaterials-13-00553-f002:**
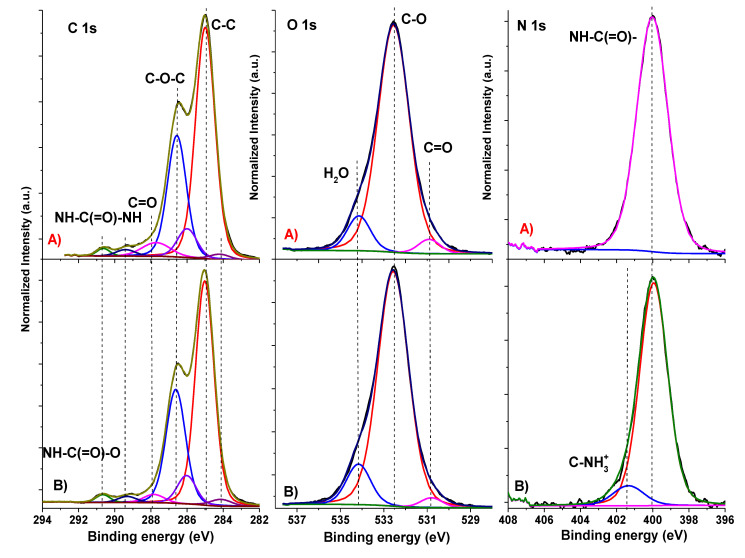
High-energy resolution XPS spectra C1s, O1s, and N1s for pure PU (**A**) and PU/1.5GO composite (**B**).

**Figure 3 nanomaterials-13-00553-f003:**
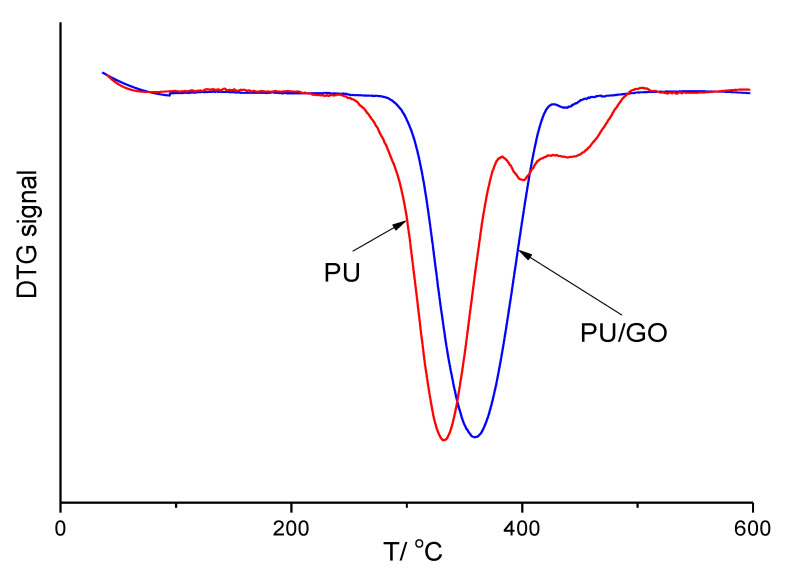
DTG curves of PU and PU/1.0GO.

**Figure 4 nanomaterials-13-00553-f004:**
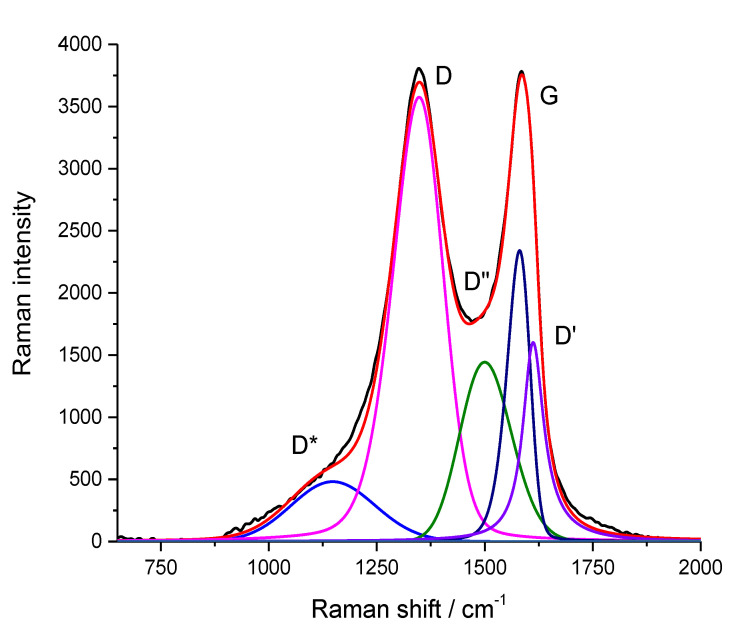
Raman spectrum of graphene oxide in the range of 750–2000 cm^−1^, along with its description by two Gaussians (peaks D* and D”) and three pseudo-Vogt functions (peaks D, G, and D’).

**Figure 5 nanomaterials-13-00553-f005:**
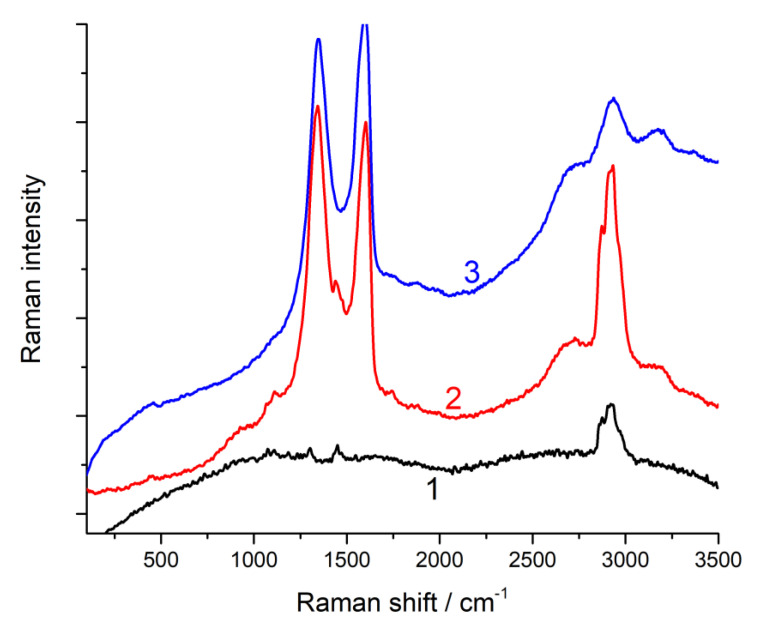
Raman spectra in the range of 50–3500 cm^−1^ of PU (1) and its composites with 1% GO (2) and 1.5% GO (3).

**Table 1 nanomaterials-13-00553-t001:** Elemental analysis of PU/GO composites.

Sample	Content, wt.%				
	C	H	N	S	O *
PU	58.03	8.74	2.99	0.29	29.96
PU/0.1GO	57.35	8.77	2.92	0.11	30.86
PU/0.5GO	57.76	8.74	2.96	0.37	30.18
PU/1.0GO	58.14	8.84	3.07	0.41	29.55
PU/1.5GO	57.89	8.77	2.94	0.19	30.21
GO	45.28	2.72	0.00	2.30	49.70

* Oxygen content was estimated by the formula [O] = 100—$\sum_{i}[C_{i}$], where [*C_i_*] is the content of the *i*-th element

**Table 2 nanomaterials-13-00553-t002:** XPS-derived content (in atomic units) and temperature of maximum degradation rate T_max_ (in ^o^C) for PU and PU/GO composites.

Sample	Content, at.%					T_max_,
	C	O	N	S	Si	°C
PU	75.87	16.71	1.97	0.11	5.34	340.5
PU/0.1GO	76.31	16.35	1.92	0.13	5.29	348.0
PU/0.5GO	76.55	16.51	1.73	0.15	5.06	366.4
PU/1.0GO	76.14	16.98	1.58	0.14	5.15	363.6
PU/1.5GO	77.29	16.42	1.50	0.09	4.71	358.1
GO	73.23	24.28	>0.30	2.19	-	-

**Table 3 nanomaterials-13-00553-t003:** Atomic ratios for PU and PU/GO composites according to Elemental analysis and XPS.

Sample	Atomic Ratios Elemental XPS			
	C/O	C/N	C/O	C/N
PU	2.58	22.7	4.54	38.5
PU/0.1GO	2.47	23.0	4.67	39.8
PU/0.5GO	2.54	22.8	4.64	44.2
PU/1.0GO	2.62	22.2	4.48	48.1
PU/1.5GO	2.55	23.0	4.70	51.5

**Table 4 nanomaterials-13-00553-t004:** Contributions of individual chemical moieties in the high-resolution C1s, N1s and O1s spectra of PU and PU/1.5GO.

	Moiety	Binding Energy [eV]	PU/1.5GO [at%]	PU [at%]
**C1s**	C=C	284.2	1.9	1.31
	C–C	285.0	56.15	55.73
	CC–OH, –NH_2_	286.0	7.11	6.93
	C–O–C	286.6	29.57	28.41
	C=O	287.8	2.27	4.87
	C=NH–(O)–NH	289.3	1.54	1.57
	C=NH–(O)–O	290.6	1.47	1.3
**N1s**	C=NH–(O)–	399.9	90.8	100
	C–NH_3_^+^	401.3	9.2	-
**O1s**	O=C	530.8	2.5	3.84
	O–C	532.5	85.89	86.43
	H_2_O	534.2	11.61	9.74

**Table 5 nanomaterials-13-00553-t005:** Comparison of the effect of nanofillers in polyurethane composites.

Polyurethane Matrix	Nanofiller	Content, wt.%	Processing Method	Thermal Stability	Highlights	Ref.
TPU	RGO	0.1	solution	Increased 6 °C	410% toughness 8% hardness	[38]
TPU	OMMT	1.0	in situ	Increased 10 °C	T_m_ increased	[39]
TPU	MWCNT	2.0	melt	Increased 13 °C	Increased modulus	[40]
PU	f-GNP	1.5	solution	Increased 30 °C	Enhanced shape memory	[41]
PU	GNS	2.0	in situ	Increased 40 °C	202% storage modulus	[42]
PU	GO	1	solution	Increased 21 °C	elongation at break 64.5%,	[43]
PU	GO	0.1	solution	Increased 7.5 °C	T_m_ increased, increase concentration of defects in structure nanosheets	Present work
PU	GO	0.5	solution	Increased 26 °C		

Abbreviations: TPU—thermoplastic polyurethane; OMMT—organically modified montmorillonite; T_m_—melting temperature; PU—polyurethane; MWCNT—multiwall carbon nanotube; f-GNP—functionalized graphene nanoplatelets; GO—graphene oxide; RGO—reduced graphene oxide; GNS—graphene nano-sheets.

**Table 6 nanomaterials-13-00553-t006:** The positions (*Pos*), full widths at half maximum (*FWHM*), and intensities (*Int*) of the peaks in the Raman spectra of the samples under study.

Sample	Peak	*Pos*, cm^–1^	*FWHM*, cm^–1^	*Int*, %	I_D_/I_G_
GO	D*	1114.2	178	3.2	
	D	1348.5	134	60.3	
	D*”*	1520.0	123	4.8	2.09
	G	1587.1	78	28.8	
	D*’*	1613.3	32	3.0	
rGO	D*	1144.2	220	5.2	
	D	1347.9	80	60.1	
	D”	1522.2	140	9.7	2.68
	G	1584.0	55	22.4	
	D’	1617.1	27	2.7	
PU/1.0GO	D*	1100.5	180	5.6	
	D	1338.7	111	55.0	
	D”	1500.0	187	21.0	7.24
	G	1580.0	71	7.6	
	D’	1605.9	56	10.8	
PU/1.5GO	D*	1150.0	144	2.3	
	D	1346.6	118	45.8	
	D”	1520.0	221	30.2	3.69
	G	1580.0	72	12.4	
	D’	1602.8	56	9.4	

## Data Availability

The data supporting reported results can be obtained on request from the authors.

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
