# Peer review of "On the State of Graphene Oxide Nanosheet in a Polyurethane Matrix"

_nanomaterials, 2023, doi:10.3390/nano13030553_

Round 1

Reviewer 1 Report

This paper describes synthesis and characterization of composites composed of aliphatic polyurethane (PU) and graphene oxide (GO).  The authors found that the introduction of GO into a PU matrix was accompanied by a partial reduction of GO nanosheet and an increase in the concentration of defects in its structure.  I think the experiments were carefully done and the characterization date are reliable.  Probably, the composites are not homogeneous mixture of the two segments, instead GO exists mainly on the surface of PO.  The results of this paper will give useful information in the field of filler containing PU materials.  I would like to accept this manuscript in nanomaterials.

May I have comments.

- Since silicone is coming from dimethylsiloxane-based contaminants, there should be some contribution of carbon content from the contaminants.

- I am just curious of the mechanism of GO reduction.

- What is the reason why PU showed existence of sulfur? (Table 1)

Author Response

Response to Reviewer 1 Comments

Reviewer 2 Report

In this paper, the authors studied the mechanical strength and thermal stability of graphene films obtained from aqueous polyurethane (PU) dispersions by elemental analysis, X-ray photoelectron spectroscopy, differential thermal gravimetry and Raman spectroscopy. It was found that the introduction of GO into a PU matrix was accom-panied by a partial reduction of graphene oxide nanosheet and an increase in the concentration of defects in its structure. I believe that publication of the manuscript may be considered only after the following issues have been resolved.

1.       In order to better highlight the advantages of this work, the author needs to provide a table to compare related work.

2.       The author of the abstract needs to rewrite to reflect the core of this article. And highlight the significance of this work.

3.       What is the morphology of graphene oxide in this work? The author needs to give some proof.

4.       The impression of this work is that some methods have been used to analyze graphene oxide, and the necessary theoretical explanation and the prospect of follow-up work are lacking. The author needs to supplement.

5.       The introduction can be improved. The articles related to some applications of graphene materials should be added such as Sensors 2022, 22, 6483; ACS Sustain. Chem. Eng. 2015, 3, 1677–1685; Diamond & Related Materials 128 (2022) 109273; Talanta 2015, 134, 435–442.

6.       Please check the grammar and spelling mistakes of the whole manuscript.

Author Response

Authors' Responses to Reviewer's Comments (Reviewer 2)

Round 2

Reviewer 2 Report

 Accept in present form.